# The Combination of Binding Avidity of Ovomucoid-Specific IgE Antibody and Specific IgG4 Antibody Can Predict Positive Outcomes of Oral Food Challenges during Stepwise Slow Oral Immunotherapy in Children with Hen’s Egg Allergy

**DOI:** 10.3390/nu15122770

**Published:** 2023-06-16

**Authors:** Shoichiro Taniuchi, Rika Sakai, Takahiro Nishida, Meguru Goma, Masatoshi Mitomori, Aya Imaide, Masahiro Enomoto, Masamitsu Nishino, Yo Okizuka, Hiroshi Kido

**Affiliations:** 1Department of Pediatrics, Takatsuki General Hospital, Osaka 569-1192, Japan; t_nishida0930@yahoo.co.jp (T.N.); megurugoma@gmail.com (M.G.); aabbk7k7k7@gmail.com (M.M.); kiku.tami.ume@gmail.com (A.I.); masahiro.enomoto@gmail.com (M.E.); m-nishino@ajk.takatsuki-hp.or.jp (M.N.); okizuka730@gmail.com (Y.O.); 2Division of Enzyme Chemistry, Institute for Enzyme Research, Tokushima University, Tokushima 770-8501, Japan; rsakai00@tokushima-u.ac.jp

**Keywords:** oral immunotherapy, hen’s egg allergy, IgE binding avidity, ovomucoid-specific IgE, ovomucoid-specific IgG4, oral food challenge

## Abstract

To increase the prediction accuracy of positive oral food challenge (OFC) outcomes during stepwise slow oral immunotherapy (SS-OIT) in children with a hen’s egg (HE) allergy, we evaluated the predictive value of the combination of antigen-specific IgE (sIgE) with antigen binding avidity and sIgG4 values. Sixty-three children with HE allergy undergoing SS-OIT were subjected to repeated OFCs with HE. We measured the ovomucoid (OVM)-sIgE by ImmunoCAP or densely carboxylated protein (DCP) microarray, sIgG4 by DCP microarray, and the binding avidity of OVM-sIgE defined as the level of 1/IC_50_ (nM) measured by competitive binding inhibition assays. The OFC was positive in 37 (59%) patients undergoing SS-OIT. Significant differences in DCP-OVM-sIgE, CAP-OVM-sIgE, IC_50_, DCP-OVM-sIgG4, the multiplication products of DCP-OVM-sIgE, and the binding avidity of DCP-OVM-sIgE (DCP-OVM-sIgE/IC_50_) and DCP-OVM-sIgE/sIgG4 were compared between the negative and positive groups (*p* < 0.01). Among them, the variable with the greatest area under the receiver operating characteristic curve was DCP-OVM-sIgE/IC_50_ (0.84), followed by DCP-OVM-sIgE/sIgG4 (0.81). DCP-OVM-sIgE/IC_50_ and DCP-OVM-sIgE/sIgG4 are potentially useful markers for the prediction of positive OFCs during HE-SS-OIT and may allow proper evaluation of the current allergic status in the healing process during HE-SS-OIT.

## 1. Introduction

Hen’s egg (HE) allergy is one of the most frequent food allergies (FA) in Western countries, as well as in Japan, and its incidence has been on the rise lately [1]. It is estimated to affect between 1.6% and 10.1% [2] and as many as 9.5% of children [3,4,5]. The onset often occurs before the first birthday [6]. The prevalence and age at acquisition of spontaneous oral tolerance in children vary among studies probably due to differences in populations, settings, and the methods used for confirming the diagnosis. Epidemiological studies in various countries estimate the prevalence of HE allergy to range from 1.2% to 2.9% at 2 years of age [7,8,9,10,11]. Natural resolution has been estimated in up to 80% of the cases at 3 years of age and in 38–90% of children at 5–6 years of age [12]. Furthermore, clinical HE hypersensitivity disappears in 60% of children aged between 6 and 12 years [12]. 

Oral immunotherapy (OIT) may improve the quality of life and raise the threshold at which a patient with FA may react to an allergen. The guidelines issued by several countries have also recommended that OIT can be used as a potential treatment in such individuals [13,14]. OIT is also commonly used for the treatment of young children with HE allergies and has been reported to be effective in several studies [15,16]. However, adverse effects are associated with the occasional use of parenteral epinephrine during OIT, particularly during the escalation phase. Almost a decade ago, our group reported the development of stepwise slow (SS)-OIT for HE allergy that can avoid adverse reactions during the escalation phase [17]. The SS-OIT can achieve desensitization by a gradual increase in the loading dose at oral food challenges (OFCs) delivered at 2–3 month intervals without the escalation phase. Nevertheless, we reported in the same study that 16 of 30 (53.3%) children in the OIT group experienced adverse reactions following OFC involving dose build-up in the hospital, and the same rate was noted for home OIT intake [17]. Therefore, it is important to predict provoked allergic reactions during OFCs during SS-OIT in order to achieve oral tolerance for each allergen. 

Determination of allergen specific-IgG4 (sIgG4) and allergen-specific IgE (sIgE)/sIgG4 ratio is currently used to predict the natural history of FAs and the response to immunotherapy [18]. ImmunoCAP (Thermo-Fisher Scientific Inc., Uppsala, Sweden) (CAP)-ovalbumin and -ovomucoid (OVM)-specific sIgE/sIgG4 ratios are higher in patients with allergy to baked HE products [18]. Conversely, an increase in the CAP-OVM-sIgG4/sIgE ratio during immunotherapy was associated with a successful clinical outcome [19]. Our group also reported previously that, in 64 patients with 105 positive oral challenges with heated HE, the area under curve (AUC) for CAP-HE-white (EW)-sIgE, CAP-EW-sIgG4, and CAP-EW-sIgE/sIgG4 for predicting positive results were 0.609, 0.724, and 0.847, respectively [20]. We concluded that the CAP-EW-sIgE/sIgG4 ratio generated significantly higher specificity, sensitivity, positive predictive value (%), and negative predictive value (%) than the individual IgE and IgG4. In addition to our study, others have reported the usefulness of OVM-sIgE levels in predicting an IgE-mediated allergy to heated HE [21], and that OVM-sIgE is superior to EW-sIgE for the diagnosis of a heated HE allergy [22].

To monitor immunoglobulin isotype production, we reported in a series of recent studies the development of a highly sensitive densely carboxylated protein (DCP) microarray with a high antigen immobilization capacity [23,24,25]. The DCP microarray offers the advantages of allowing the measurement of various immunoglobulin isotypes, such as IgG4 and IgE, under the same assay conditions using a small volume of serum or plasma (100 μL for all analyses). In addition, we also recently reported that a DCP microarray can detect allergen-specific low- and high-affinity IgE isotypes and high-affinity IgE, but not low-affinity IgE, which can elicit mast cell degranulation [26]. In our most recent study [27], we reported that the combination of the quality (binding avidity, 1/IC_50_) and quantity (level) of OVM-sIgE [OVM-sIgE/IC_50_] provides a more accurate prediction of heated HE allergy in young children with low OVM-sIgE levels, compared with a OVM-sIgE level alone. However, that study used CAP-OVM-sIgE, not DCP-OVM-sIgE measurements. Taking the above findings together, we hypothesized that the two parameters DCP-OVM-sIgE/IC_50_ and DCP-OVM-sIgE/IgG4 obtained using a DCP microarray may be suitable biomarkers for monitoring allergic symptoms, such as a positive OFC during HE-OIT. To test this hypothesis, we used the novel biomarkers of DCP-OVM-sIgE/IC_50_ and DCP-OVM-sIgE/sIgG4. Here, we assessed the value of the combination of DCP-OVM-sIgE/IC_50_ and DCP-OVM-sIgE/IgG4 in predicting allergic reactions at repeated OFCs following the ingestion of heated HE allergen during long-term HE-OIT.

## 2. Materials and Methods

### 2.1. Study Design

An exploratory prospective cohort study was conducted from March 2020 to February 2023 at the Department of Pediatrics, Takatsuki General Hospital, Osaka, Japan.

### 2.2. Patients

The criteria for inclusion in this study were as follows: (i) age 6–15 years, and (ii) receiving OIT for HE-FA. The patients were already diagnosed with HE-FA by HE-OFC and their parents provided written informed consent for the OIT protocol. The following exclusion criteria were applied at study entry: (i) severe atopic dermatitis [28] and (ii) uncontrolled asthma [baseline FEV1 (forced expiratory volume one second) <80% of predicted value] in accordance with the Japanese Pediatric Guidelines for the treatment and management of bronchial asthma [29]. The patients were recruited by these criteria and written informed consent was obtained at the start of the study.

### 2.3. Oral Food Challenge (OFC)

An OFC was performed on admission to our hospital, for the diagnosis of HE-FA before the commencement of OIT. An OFC was conducted at 1 h intervals using a graded dosing method and performed under the supervision of a physician. Basically, patients ingested heated whole HE, starting at 1 g, followed at 1 h intervals by increased doses (2 g, 5 g, 10 g). This approach was, however, not used in patients considered at risk of reacting to minimum doses. These patients instead received a lower dose using HE powder (Tamakona 250^®^; Tamakona Inc., Kobe, Japan); the starting dose of heated HE powder was 25 mg, which is equivalent to 0.2 g of whole HE boiled for 15 min, followed 1 h later by 100 mg. The OFC was performed in a hospital setting under supervision of a physician, though subjective symptoms (nausea, sore throat, or itching) were not assessed. When present, eczema and/or asthma were well-controlled in all children before they underwent OFCs. For preparation of the heated HE, the whole HE was cooked and then steamed for 15 min.

### 2.4. SS-OIT with Repeated OFCs

Written informed consent was obtained at the start of the SS-OIT protocol for FA, which was approved by the institutional ethics committee of Takatsuki General Hospital (#2017-71). We used the SS-OIT as described previously [17] with some modification. Figure 1 illustrates the SS-OIT schedule used in this study. The patient was admitted to our hospital for an OFC during SS-OIT. The dosing frequency was once per OFC. The initial dose of SS-OIT was 12.5 mg of HE protein (Tamakona 250^®^), followed by 25, 50, and 250 mg, and then switched to 3 g of heated whole HE, followed by 5, 10, 15, 20, 25, 30, 35, 40, and 50 g. The patient ingested heated HE powder or heated whole HE at the time of each OFC and 2 h later if no provoked allergic symptoms were noted. They were then discharged and advised to continue consuming the safe dose of HE materials (HE powder or heated whole HE) three times a week at home. The challenge of ingesting an increased dose of heated HE powder or heated whole HE was conducted every 3 months under medical supervision at follow-up visits. When allergic symptoms appeared during an OFC at home, the patient was advised to continue using the previously safe dose.

The escalation phase was repeated in the same manner until the child was able to ingest 50 g of HE, corresponding to a medium-sized HE. This was defined as the minimum dose required to achieve the induction of oral tolerance to one of the most popular Japanese food preparations. When severe allergic symptoms were expected during repeated OFC based on history of anaphylaxis and the level of OVM-sIgE, the dose increase rate at the next OFC was reduced or the interval until the next OFC was extended. We also changed the provided sample to baked HE products (e.g., muffins or pancakes).

### 2.5. Definition of Positive OFC during HE-SS-OIT

Positive OFC was defined as the appearance of allergic symptoms of higher than grade 1 at repeated OFCs, in accordance with Japanese food allergy guidelines [30] at least once during the period between 6 months before and 6 months after the date of blood sampling (Figure 1). Only objective symptoms were used on the judgment of the OFC. If a clear judgment of OFC outcome was not possible because of very mild allergic symptoms, such as nausea, sore throat, or itching, we judged it on the next visit after an OFC, based on the symptoms at home upon repeated intake of the same dose as in the OFC. We divided the subjects into two groups: those positive and negative at repeated OFCs during OIT. The allergic symptoms were judged only at the OFC, except for those emerging accidentally at home. If no positive OFC occurred during the period, data from the OFC performed nearest the sampling date were used for the analysis. In addition, if positive OFCs occurred more than once upon repeated OFCs, we used the data of loading dose from the OFC when the strongest positive OFC occurred.

### 2.6. Measurements of Total IgE, CAP-OVM-sIgE, DCP-OVM-sIgE, 1/IC_50_, and DCP sIgG4

Blood sampling was usually undertaken once a year during the SS-OIT. Serum samples were stored at −70 °C until analysis. Total IgE and OVM-sIgE levels were measured by ImmunoCAP. DCP-OVM-sIgE and DCP-OVM-sIgG4 were measured by DCP microarrays. The binding avidity of OVM-sIgE was measured as the competitive binding inhibition activity between OVM immobilized on a DCP microarray and serially diluted soluble OVM and was defined as the level of 1/IC_50_ (nM) [23,24,25,26]. IC_50_ represents the concentration of allergen required for 50% binding inhibition.

Before analysis, the level of anti-OVM-sIgE in each serum sample bound on the DCP microarray was adjusted to a fluorescence intensity of 2000 by the dilution buffer of phosphate-buffered saline (PBS) containing 0.3 M KCl, and 0.05% Tween 20. Soluble OVM at 28 mg/mL (1 mM) was adjusted to final concentrations of 1, 5, 20, and 50 nM by the dilution buffer, added to the serum sample followed by incubation at 25 °C for 30 min. After incubation, each reaction mixture was applied to an OVM-immobilized DCP microarray, and a competitive binding inhibition assay was conducted at 37 °C for 120 min. After washing three times, the bound antibody was detected with a HiLyte Fluor 555-labeled secondary antibody against human IgE, followed by the calculation of IC_50_. We reported previously the intra- and inter-assay coefficients of variation for the DCP microarray assay as follows: inter-assay: 5.70% to 13.5%, within-slide: 7.70% to 25.2%; and batch-to-batch: 2.7% to 24.4% [27]. The measurements of DCP-OVM-sIgE, 1/IC_50_, and DCP sIgG4 were performed by blinded researchers in the Division of Enzyme Chemistry at the Institute for Enzyme Research, Tokushima University, from March 2020 to March 2023.

### 2.7. Statistical Analysis

The Mann–Whitney U test and χ^2^ test (two-tailed), including Fisher’s exact test, were used to test for the significance of differences between groups. Performance characteristics (i.e., sensitivity and specificity) were calculated for various cut-off values, including the optimal cut-off values proposed based on the receiver operating characteristic (ROC) plots. Logistic regression was used to identify clinical and mechanistic factors (log-transformed where noted) that might identify positive OFC. All statistical analyses were performed with EZR (easy R) [31], which is a modified version of the R commander designed to add statistical functions frequently used by biostatisticians.

## 3. Results

### 3.1. Patient Selection

The patient selection process is shown in Figure 2. A total of 96 patients met the inclusion criteria before the application of the exclusion criteria. Between March 2020 and February 2023, the parents of each patient provided a signed informed consent for SS-OIT with an HE allergy. Furthermore, 33 patients were excluded [no serum was obtained for analysis (*n* = 13), receiving treatment for OIT for egg yolk (*n* = 2) and raw egg (*n* = 8), and received blood sampling twice during the study (*n* = 10)]. Thus, the study population included 63 patients [46 (73%) males; median age at testing, 10 years, range 8.7–11.2 years].

### 3.2. Patients’ Characteristics

Table 1 summarizes the clinical characteristics of the 63 patients of the HE-OIT group at the start of OIT and at blood sampling. Table 2 provides a comparison of the clinical characteristics of the negative and positive groups. No significant differences were noted between the two groups in any of the following baseline characteristics (median values): age (negative vs. positive group, 9.7 years vs. 10.5 years), history of bronchial asthma, history of atopic dermatitis, history of allergic rhinitis, history of anaphylaxis, sex, frequency of consumption of baked HE products, the period since the start of OIT (28.5 vs. 32.0 months), the threshold dose of heated HE of OFC at the start of SS-OIT (2.0 vs. 1.0 g), total number of OFC before the sampling date (7 vs. 6 OFCs), and total IgE titer at the date of blood sampling (922 vs. 934 IU/mL). The loading dose of HE at the OFC of the negative group was significantly higher than that of the positive group (18 vs. 6 g, *p* = 0.0271). These data were analyzed using Fisher’s exact test or Mann–Whitney U test for categorical and continuous variables, respectively.

### 3.3. Comparisons of Immunological Variables for Discriminating Positive and Negative Groups at Repeated OFCs during SS-OIT 

The six serum IgE-related parameters are as follows: CAP-OVM-sIgE (UA/mL), DCP-OVM-sIgE (BUe/mL), binding avidity of OVM-sIgE (1/IC_50_) (nM), DCP-OVM-sIgG4 (BUg4/mL), the product of DCP-OVM-sIgE multiplied by binding avidity of DCP-sIgE (DCP-OVM-sIgE/IC_50_), and the product of DCP-sIgE divided by DCP-sIgG4 (DCP-OVM-sIgE/sIgG4) were all significantly higher in the positive group than in the negative group (Table 3 and Figure 3, by Mann–Whitney U test). Among the parameters tested, DCP-sIgE/IC_50_ and DCP-sIgE/sIgG4 demonstrated good accuracy (AUC ≥ 0.81) with sensitivity of ≥70% and specificity ≥78%, although DCP-OVM-sIgE and CAP-OVM-sIgE by themselves showed good accuracy (AUC 0.7–0.8) for predicting positive OFC during SS-OIT (Figure 4). Receiver operating characteristic (ROC) analysis demonstrated a larger area under curve (AUC) for DCP-OVM-sIgE/IC_50_ compared with DCP-OVM-sIgE alone (*p* = 0.03) and also DCP-sIgG4 (*p* = 0.04). There was no significant difference in the level of AUC among the other parameters (Figure 4). We also determined the adjusted odds ratios of these six parameters by multiple logistic analysis using two factors (age and loading dose of HE) as covariates (Table 4). The highest adjusted odds ratios were observed for DCP-OVM-sIgE/IC_50_ and DCP-sIgE/sIgG4 (15.6 and 8.6, respectively). Considered together, the two parameters of DCP-OVM-sIgE/IC_50_ and DCP-sIgE/sIgG4 seem to accurately predict positive OFC during OIT.

### 3.4. Two-Dimensional Heat Map Analysis Using DCP-OVM-sIgE/IC50 and DCP-OVM-sIgE/IgG4

A two-dimensional heat map analysis using DCP-OVM-sIgE/IC_50_ and DCP-OVM-sIgE/IgG4 parameters was performed to discriminate between cases positive and negative for positivity of repeated OFCs. The cut-off values for positive and negative OFCs calculated from ROC analysis are shown on the abscissas and ordinates of the plots shown in Figure 5A. Positive (red) and negative (blue) cases could be clearly discriminated using the indicated cut-off values. However, there was no clear discrimination between positive and negative groups using the combination of DCP-OVM-sIgE and DCP-sIgE/IC_50_, or DCP-OVM-sIgE and DCP-OVM-sIgE/IgG4. As shown in Figure 5B, the two-dimensional graph including both DCP-OVM-sIgE/IC_50_ and DCP-OVM-sIgE/IgG4 values showed that the condition of each patient was distributed in one regression line, along with the 95% confidence interval (R = 0.802, 95% CI: 0.691–0.876, *p* < 0.001, by Pearson’s correlation test).

To examine the quantitative discrimination ability in predicting positive OFCs using the combination of the two parameters in the two-dimensional heat map analysis, 63 children were divided into four subgroups using cut-off values of these two parameters. Positive and negative predictive values were determined, as well as odds ratios using the combination of the cut-off values of these two parameters between the negative and positive groups among the four subgroups (Table 5). The two parameters (DCP-OVM-sIgE/IC_50_, DCP-OVM-sIgE/sIgG4) were used for categorization into four subgroups as follows: group A, 7.15 or more and 3.48 or more; group B, 7.15 or more and less than 3.48; group C, less than 7.15 and 3.48 or more; and group D (reference group), less than 7.15 and less than 3.48, respectively.

A multiple logistic regression analysis was performed to compare group D with groups A, B, and C using all potential confounder factors as covariates (Table 5). The highest adjusted odds ratios were observed for group A and group B: 50.6 and 12.0 with significant values of C-statistics of 0.88 and 0.82, respectively. The positive predictive value and the negative predictive value were extremely high (0.88, 0.88) in group A and group D, respectively. Taken together, the two parameters DCP-OVM-sIgE/IC_50_ and DCP-sIgE/sIgG4 appeared to accurately predict positivity for OFC during OIT.

## 4. Discussion

In this study, we evaluated the predictive value of baseline characteristics and a series of OVM-specific antibodies in particular to elucidate the risk of allergic symptoms associated with positive OFC during HE-SS-OIT.

Various OIT methods have been reported which employ different durations of OIT, dosing, and process of food heating. The typical OIT protocol includes the following three steps [32]: a rapid escalation phase, a build-up phase, and a maintenance phase. Generally, hospital admission is necessary during the escalation phase to identify/treat potentially severe allergic reactions. In the HE-SS-OIT study, certain allergic reactions were of sufficient clinical significance that some (approximately 15%) of the children who received OIT did not complete it [33]. In most cases, this was due to allergic reactions in the first few months during OIT [33]. Several modifications have been introduced to avoid adverse events during OIT. For example, a recent study [34] reported comparable therapeutic efficacy of a reduced maintenance dose of HE OIT compared with the target dose. The same study also concluded that reducing the maintenance dose for HE, milk, and wheat seems to effectively lower the severity of symptoms associated with their consumption compared with that at the target OIT dose [34]. Our group also reported that SS-OIT for the HE allergy without a rapid escalation phase was effective in achieving desensitization to the HE allergy, compared with the findings in an untreated group [17]. However, 16 of 30 (53.3%) children of the HE-SS-OIT group experienced adverse reactions following in-hospital challenges involving dose build-up, and the same rate was true for home OIT intake [17]. Taking these findings together, there is a need for novel biomarkers that can monitor allergic reactions of patients more precisely at OFC and indicate the position in the overall healing process of allergy during SS-OIT. 

We have recently reported the significance of measuring the binding avidity of OVM-sIgE (OVM-sIgE/IC_50_) for the diagnosis of heated HE allergy in young children with low OVM-sIgE levels, compared with the quantity of OVM-sIgE alone [27]. In the present study, the binding avidity of OVM-sIgE (OVM-sIgE/IC_50_) showed good accuracy (AUC 0.84), but did not reach the AUC level of 0.90 reported in the previous study [27]. This difference between these two studies may be explained as follows: (1) The number of loadings at the OFC was usually only once. Therefore, the patient did not reach the threshold dose of heated whole HE. (2) Alternatively, in the present study, DCP chips were used, while the ImmunoCAP system was used in the previous study [27]. In each measurement system, the quality of the OVM allergen used may not be the same.

Only the loading dose was significantly different between the positive and negative groups. There were no significant differences in other characteristics, although significant differences in the six immunological parameters were found. The loading dose in the positive group was less than that in the negative group at repeated OFCs during OIT. This may be dependent on the severity of allergic symptoms in each patient. 

A recent study described allergen-immunoglobulin class-switching recombination (μ → γ3 → γ1 → γ2 → γ4, and γ1 → high antigen binding affinity IgE) occurring during the process of desensitization and sensitization in patients with FA [35]. Briefly, the report suggested that oral food allergen sensitization in the early phase of infancy may cause a higher level of allergen-specific IgG1 with a detectable level IgG2 and low-antigen binding affinity IgE production, but not IgG4, which probably plays a crucial role in the development of physiological oral tolerance [35]. Alternatively, several studies have demonstrated increases in allergen-specific IgG4 during the maintenance phase of OIT, but no data of low-antigen binding affinity sIgE during OIT were reported [36]. Our study showed that the markers DCP-OVM-sIgE/IC_50_ and DCP-OVM-sIgE/sIgG4 showed the best adjusted odds ratios (15.6 and 8.6, respectively) and good accuracy (AUC 0.84 and 0.81, respectively) (Table 4 and Figure 4). In addition, the correlation coefficient of both markers was 0.804 and significant (*p* < 0.001). The two-dimensional graphs shown in Figure 5, with DCP-OVM-sIgE/IC_50_ on the abscissa and DCP-OVM-sIgE/sIgG4 on the ordinate, showed that the immunological parameters of each patient are distributed on a regression line, along with the 95% confidence interval (Figure 5B), and the allergic condition of each patient could be visualized and clearly indicated the standing position in the overall allergy status and healing process during SS-OIT. The two-dimensional heat map analysis identified three groups with different allergic conditions (encircled by three lines: the red circle line, allergy positive; red dotted circle line, allergy positive and negative mixed; and blue circle line, allergy negative, Figure 5A). The positive group was identified in early phase of OIT, the negative group in the late phase of OIT, and the mixed group in the transitional phase of OIT. This may be explained by the immunological status of patients moving from sensitization to desensitization to the allergen of OVM during SS-OIT. When responding well to SS-OIT, the immunological parameters of such patients moved leftward on the abscissa with increasing IgG4, and downward on the ordinate with increasing low-antigen binding avidity. Thus, these two parameters may be useful for monitoring the positivity of repeated OFCs as well as allergic reactions at home by ingesting allergens during healing in the process of SS-OIT. 

To quantify and evaluate the usefulness of the two-dimensional map, we compared the odds ratios in four subgroups established using cut-off values of the two parameters. Significant adjusted odds ratios were obtained in two groups (groups A and B) compared with group D (reference group). In particular, an extremely high adjusted odds ratio (50.6) was observed in association group A and group D. This result reflects the excellent accuracy of the C-statistic (0.88). In addition, a higher positive predictive value (0.88) and a higher negative predictive value (0.88) were observed in group A and group D, respectively. These results imply that the combination of the two parameters is extremely valuable in predicting OFC outcomes compared with each OVM-IgE-related parameter alone and may be useful in predicting OFCs for all ages and by other foods, as well as allergic reactions at home upon ingesting allergens during healing in the process of SS-OIT. 

Our study has several limitations. First, we performed an open OFC, but subjective symptoms were excluded. Second, the OIT study usually used a heated whole HE, but some of the children preferred to consume a baked HE products. However, positivity in the repeated OFC was not affected by using heated whole or baked HE products. Third, the rate of increase in the dose at each OFC was sometimes different. However, this study is meaningful for deciding on the increase in dose and the interval of each OFC during the SS-OIT. Fourth, the study did not evaluate the correlation between severity (anaphylaxis) and OVM-related immunological parameters because only two patients developed anaphylaxis after repeated OFCs.

## 5. Conclusions

The present study is the first to report the role of binding avidity of allergen-specific IgE, which was also influenced by allergen-specific IgG4, in the desensitization achieved by SS-OIT. In addition, we demonstrated that the combination of two immunological parameters (DCP-OVM-sIgE/IC_50_ and DCP-OVM-sIgE/IgG4) is potentially useful for monitoring allergic adverse events at repeated OFCs and may reflect the allergic status during HE-SS-OIT. These parameters could also be significant in deciding the rate of increase in exposure to OFC material in each OFC during SS-OIT.

## Figures and Tables

**Figure 1 nutrients-15-02770-f001:**
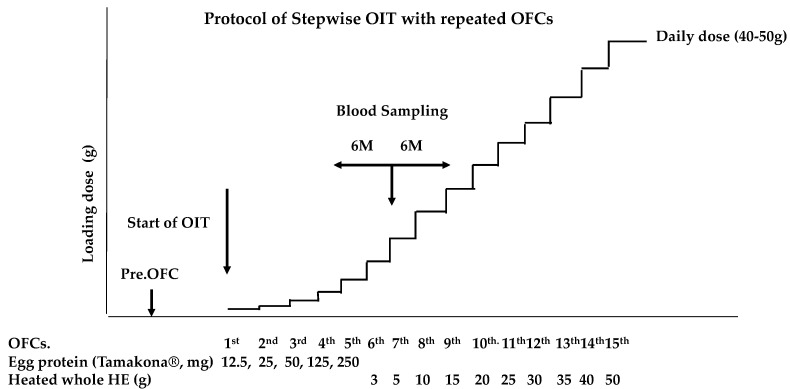
Protocol of slow stepwise OIT (SS-OIT). OFCs: oral food challenges; Pre.OFC: preliminary OFC was performed before start of the OIT. The initiation dose of the OIT is 12.5 mg of egg protein using heated HE powder (Tamakona 250^®^).

**Figure 2 nutrients-15-02770-f002:**
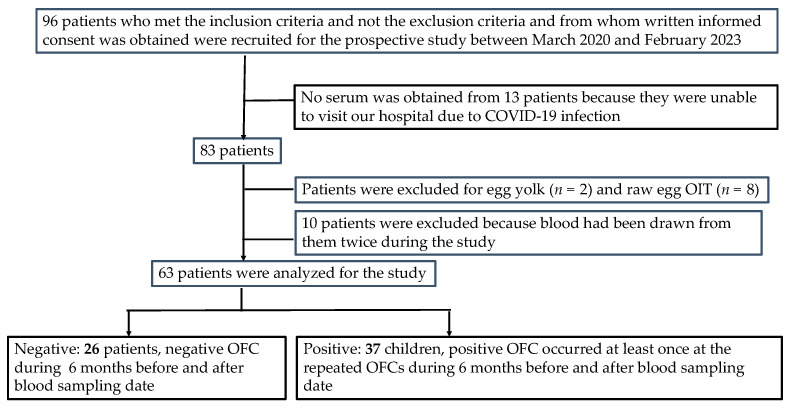
Patient selection protocol.

**Figure 3 nutrients-15-02770-f003:**
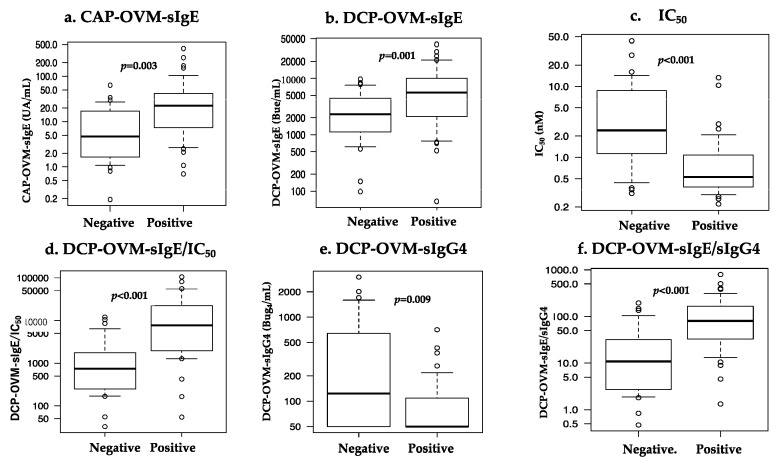
Comparison of six parameters [(**a**) CAP-OVM-sIgE (UA/mL), (**b**) DCP-OVM-sIgE (BUe/mL), (**c**) OVM-sIgE (IC_50_) (nM), (**d**) the product of DCP-OVM-sIgE multiplied by binding avidity of DCP-sIgE (DCP-OVM-sIgE/IC_50_), (**e**) DCP-OVM-sIgG4 (BUg4/mL), (**f**) the product of DCP-sIgE divided by DCP-sIgG4 (DCP-OVM-sIgE/sIgG4)] in discriminating positive from negative groups at repeated OFCs during OIT.

**Figure 4 nutrients-15-02770-f004:**
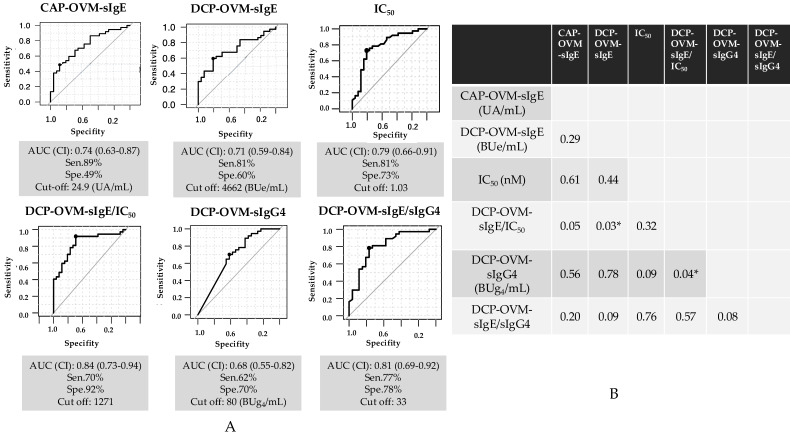
ROC curve analysis for the prediction of HE allergy at repeated OFCs during SS-OIT. (**A**) Results using six parameters: CAP-OVM-sIgE (UA/mL), DCP-OVM-sIgE (BUe/mL), OVM-sIgE (IC_50_) (nM), the product of DCP-OVM-sIgE multiplied by avidity of DCP-sIgE (DCP-OVM-sIgE/IC_50_), DCP-OVM-sIgG4 (BUg4/mL), the product of DCP-sIgE divided by DCP-sIgG4 (DCP-OVM-sIgE/sIgG4) in discriminating positive from negative groups at repeated OFCs during OIT. (**B**) Comparison of AUCs for *p* value. * *p* < 0.05. Sen: sensitivity, Spe: specificity.

**Figure 5 nutrients-15-02770-f005:**
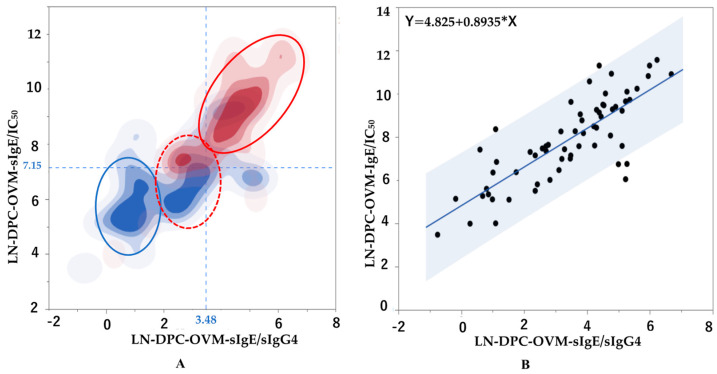
Correlations between two immunological parameters indicative of the patient’s allergic status during SS-OIT in children with HE allergy. (**A**) Heat map analysis of two parameters (abscissa: ln DCP-OVM-sIgE/IgG4, ordinate: ln DCP-OVM-sIgE/IC_50_). Horizontal blue dotted line: cut-off level of 3.48 of ln DCP-OVM-sIgE/IgG4. Vertical blue dotted line: cut-off level of 7.15 of ln DCP-OVM-sIgE/IC_50_. Red circle line: positive group at OFC, blue circle line: negative group at OFC, red dotted circle line: positive and negative groups mixed at OFC. (**B**) Two-dimensional graph including measurements of both ln DCP-OVM-sIgE/IC_50_ (abscissa) and ln DCP-OVM-sIgE/sIgG4 (ordinate) showing that the condition of each patient was located on a regression line, along with the 95% confidence interval (gray; R = 0.802, 95% CI: 0.691–0.876, *p* < 0.001 by Pearson’s correlation test).

**Table 1 nutrients-15-02770-t001:** Characteristics of 63 patients.

Age (years), median (IQR) *	7.0 (6–9)
Age (years), median (IQR) **	10.0 (8.7–11.2)
Males, *n* (%)	46 (73%)
Duration of OIT (months), median (IQR) **	30 (10–46)
Total IgE (IU/mL), median (IQR) *	891 (439–1873)
Total IgE (IU/mL), median (IQR) **	934 (525–1670)
Specific CAP OVM IgE (UA/mL) *	28.5 (11.6-76.2)
Specific CAP OVM IgE (UA/mL) **	12.6 (3.0–32.9)
History of and current allergic diseases	
Bronchial asthma, n (%)	27 (43%)
Allergic rhinitis, n (%)	35 (56%)
Atopic dermatitis, n (%)	54 (86%)
History of anaphylaxis by egg products, n (%)	23 (37%)
Characteristic at the OFC at the start of and during OIT	
Threshold dose *** (g), median (IQR) at OFC before the start of OIT	2 (0.2–5)
Loading dose *** (g), median (IQR) at each OFC during OIT	7.5 (2.3–28.5)
Positive result of repeated OFC, *n* (%) during OIT	37 (59%)
Grade 1, *n* (%)	32 (51%)
Grade 2, *n* (%)	2 (3%)
Grade 3, *n* (%)	3 (2%)
Product of HE at OFC during OIT	
Heated whole HE	39 (62%)
Baked HE product ^#^	24 (38%)
Total numbers of OFC, median (IQR) ^##^	7 (4–9)
Period (M) of OIT, median (IQR)	29 (10–40)

*: at the start of OIT **: at the date of blood sampling. ***: Dose was calculated as the weight of heated whole HE (250 mg egg protein, equivalent to 2 g of a heated whole HE) was used. ^#^: Baked HE product was prepared using whole HE (250 mg HE protein) mixed with some wheat powder. ^##^: Total numbers of OFC were calculated before the blood sampling date except preliminary OFC, OIT: oral immunotherapy, IQR: interquartile range, OVM: ovomucoid, OFC: oral food challenge, M: months OIT: oral immunotherapy, IQR: interquartile range, OVM: ovomucoid, HE: hen’s egg.

**Table 2 nutrients-15-02770-t002:** Comparison of clinical characteristics of the negative and positive groups.

	Negative Group (*n* = 26)	Positive Group (*n* = 37)	*p* Value
Age at blood sampling (years)	9.7 (8.5–10.8)	10.5 (9.1–12.0)	0.066
Males, *n* (%)	21 (81%)	25 (68%)	0.388
History of and current allergic disease			
Asthma, *n* (%)	13 (50%)	14 (39%)	0.488
Atopic dermatitis, *n* (%)	21 (82%)	33 (89%)	0.469
Allergic rhinitis, *n* (%)	16 (76%)	19 (51%)	0.587
History of anaphylaxis by egg products, *n* (%)	13 (50%)	20 (54%)	0.951
Duration of OIT (months), median (IQR)	28.5 (19.8–40.8)	32.0 (6.0–43)	0.716
Loading dose ^#^ (g), median (IQR) at each OFC during OIT	18 (4.25–39)	6 (1.8–20)	0.027
Product of HE at OFC Heated whole HE, *n* (%) Baked ^##^ HE product, *n* (%)	15 (58%) 11 (42%)	24 (64%) 13 (36%)	0.754
Threshold dose of heated whole HE ^#^ (g), median (IQR) at the start of OIT	2.0 (0.275–5.0)	1.0 (0.2–3.25)	0.082
Total number of OFC, median (IQR) until blood sampling	7 (5–9)	6 (4–9)	0.461
Total IgE (IU/mL), median (IQR) at blood sampling	922 (490–1385)	934 (671–1880)	0.312

^#^: Loading dose was calculated as the weight of whole HE. ^##^: Baked HE product was prepared using a whole HE (250 mg HE protein, equivalent to 2 g of heated whole HE) mixed with some wheat powder. See Table 1 for abbreviations. IQR: interquartile range.

**Table 3 nutrients-15-02770-t003:** Comparison of six parameters between the negative and positive groups.

	Negative Group (*n* = 26)	Positive Group (*n* = 37)	*p* Value
CAP-OVM-sIgE (UA/mL)	4.665 (1.8–16.4)	22.5 (7.4–41.6)	0.001
DCP-OVM-sIgE (BUe/mL)	2887 (1133–4371)	5305 (1930–9926)	0.003
IC_50_ (nM)	2.42 (1.16–7.58)	0.53 (0.38–1.08)	<0.001
DCP-OVM-sIgE/IC_50_	746 (253–1734)	7665 (1952–22,328)	<0.001
DCP-OVM-sIgG4 (BUg4/mL)	123 (50–579)	50.0 (50–109)	0.009
DC-OVM-sIgE/sIgG4	10.9 (2.7–30.1)	80.3 (32.7–166.4)	<0.001

All measurements are expressed as interquartile range (IQR).

**Table 4 nutrients-15-02770-t004:** Comparison of adjusted odds ratio of each parameter between the negative and positive groups.

	Adjusted Odds Ratio	95% CI	Cut-Off Value	*p* Value
CAP-OVM-sIgE (UA/mL)	6.5	2.31–28.20	24.9	0.007
DCP-OVM-sIgE (BUe/mL)	8.1	1.66–25.50	4661.8	0.001
IC_50_ (nM)	5.8	1.67–20.16	1.03	0.006
DCP-OVM-sIgE/IC_50_	15.6	4.18–58.10	1271.0	<0.001
DCP-OVM-sIgG4 (BUg_4_/mL)	0.5	0.19–1.51	79.7	0.240
DCP-OVM-sIgE/sIgG4	8.6	2.65–27.50	32.5	<0.001

Adjusted odds ratios were analyzed by logistic regression using age and dose at loading dose of OFC as covariates (each variable was divided into two using the median value).

**Table 5 nutrients-15-02770-t005:** Positive and negative predictive values and odds ratios using the combination of the cut-off values of DCP-OVM-sIgE/IC_50_ and DCP-OVM-sIgE/sIgG4 between the negative and positive groups among the four subgroups.

Name of Subgroup (*n*, %)	PPV (*n* = Positive Group)	NPV (*n* = Negative Group)	Crude Odds Ratio	95% CI	C-Statistics (AUC) ***	Adjusted Odds Ratio	95% CI	C-Statistics (AUC) ***
A (32, 51%) ^#^	0.88 (28)	0.12 (4)	54.4	8.9–332	0.86	50.6	8.2–314	0.88
B (10, 16%) ^##^	0.60 (6)	0.40 (4)	8.4	1.2–118	0.74	12.0	1.56–92	0.82
C (4, 6%) *	0.25 (1)	0.75 (3)	2.3	0.2–73.2	0.56	3.58	0.2–73.2	0.70
D (17, 27%) **	0.12 (2)	0.88 (15)						

^#^: DCP-OVM-sIgE/sIgG4 of 3.48 or more and DCP-OVM-sIgE/IC_50_ of 7.15 or more; ^##^: DCP-OVM-sIgE/sIgG4 less than 3.48 and DCP-OVM-sIgE/IC_50_ of 7.15 or more; *: DCP-OVM-sIgE/sIgG4 of 3.48 or more and DCP-OVM-sIgE/IC_50_ less than of 7.15; **: (reference group) DCP-OVM-sIgE/sIgG4 of less than 3.48 and DCP-OVM-sIgE/IC_50_ of less than 7.15. Adjusted odds ratios were analyzed by logistic regression using all confounding factors (each variable was divided into two using the median value). PPV: positive predictive value; NPV: negative predictive value; CI: confidence interval; ***: C-statistic means the value of AUC (area under the curve).

## Data Availability

Data presented in this study are available on request from the corresponding author.

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
