# Peer review of "The Combination of Binding Avidity of Ovomucoid-Specific IgE Antibody and Specific IgG4 Antibody Can Predict Positive Outcomes of Oral Food Challenges during Stepwise Slow Oral Immunotherapy in Children with Hen’s Egg Allergy"

_nutrients, 2023, doi:10.3390/nu15122770_

Round 1
Reviewer 1 Report
The authors present an exploratory prospective cohort study that aims to evaluate the predictive value of the combination of antigen-specific IgE (sIgE) with antigen binding-avidity and sIgG4 values. Sixty-three children with HE allergy undergoing stepwise slow oral immunotherapy (SS-OIT) were subjected to repeated oral food challenge (OFC) with hen’s egg (HE).
Comments:
1. The study is interesting and the results are imported.
2.
Figure 5 A Heatmap analysis of two parameters (abscissa: ln DCP-OVM-sIgE/IgG4, ordinate: ln DCP-OVM-sIgE/IC50).
Horizontal blue dotted line: cut-off level of 3.48 of ln DCP-OVM-sIgE/IgG4.
Vertical blue dotted line: cut-off level of 7.15 of ln DCP-OVM-sIgE/IC50.
Red circle line: positive group at OFC,
blue circle line: negative group at OFC,
red dotted circle line: positive and negative groups mixed at OFC.
If a cut-off level of 3.48 of ln DCP-OVM-sIgE/IgG4 and a cut-off level of 7.15 of ln DCP-OVM-sIgE/IC50 are predictive values, the joint (combined) effects should be predicted the outcome of a positive oral food challenge (OFC).
Suggested Table:
1) A new Table (2 factors, joint (combined) effects) should be created.
2) The 2 factors can be categorized into 4 groups <3.48 and <7.15 (reference group), <3.48 and ≥7.15, ≥3.48 and <7.15, and ≥3.48 and ≥7.15 between positive and negative oral food challenge (OFC) groups.
3) The sample size and percentage (number (%)) of each group should be presented in Table.
4) Crude and adjusted odds ratio with their 95% CI by using the logistic regression model should be calculated to determine the strength and presence of association.
5) Several potential confounders in Table 2 should be included in the adjusted logistic regression model.
6) The effect size (odds ratio with 95% CI) and c-statistic should be presented in the Table.
7) The c-statistic should be presented.
8) The positive predictive value might be presented in the subgroup, particularly in the subgroup ≥3.48/≥7.15.
Reviewer 2 Report
The authors report on the binding avidity of allergen-specific IgE.
Minor comments for authors:
Do the authors have data on other allergen specific IgG subclasses and IgA?
Do the authors have data on changes in FcER1 receptor levels on the basophils of these individuals?
Do the authors have data on basophil counts in these individuals?
Grammar should be double checked, especially in Figure 2. Some of the descriptors in the protocol are not clear ("96 patients met inclusion criteria but not exclusion criteria were recruited" and "no serum of 13 patients was obtained" and 10 patients were excluded for two blood samplings")
Round 2
Reviewer 1 Report
Minor Comments:
Please double-check the numbers or sample sizes in Table 5.
PPV: 28+6+1+2=37= Positive group (n=37)
NPV: 4+4+3+16=27> Negative group (n=26)
PPV (n=positive OFCs or n= Positive group),
NPV (n=positive OFCs or n= Negative group)??
D (n=17, 27%), NPV=0.89 (n=16), 16/17=0.94??
